# The IL-17A/IL-17RA Axis Is Not Related to Overall Survival and Cancer Stem Cell Modulation in Pancreatic Cancer

**DOI:** 10.3390/ijms21062215

**Published:** 2020-03-23

**Authors:** Jiahui Li, Christopher Betzler, Philipp Lohneis, Marie Christine Popp, Jiwei Qin, Thomas Kalinski, Thomas Wartmann, Christiane J. Bruns, Yue Zhao, Felix C. Popp

**Affiliations:** 1Department of General, Visceral, Tumor and Transplantation Surgery, University Hospital of Cologne, Kerpener Straße 62, 50937 Cologne, Germany; jiahui.li@uk-koeln.de (J.L.); chrisbetzler@web.de (C.B.); marie.popp@uk-koeln.de (M.C.P.); christiane.bruns@uk-koeln.de (C.J.B.); 2Institute of Pathology, University Hospital of Cologne, Kerpener Straße 62, 50937 Cologne, Germany; philipp.lohneis@uk-koeln.de; 3Department of General, Visceral und Vascular Surgery, Otto von Guericke University Magdeburg, Leipziger Str. 44, 39120 Magdeburg, Germany; jiweiqin1120@hotmail.com (J.Q.); Thomas.Wartmann@med.ovgu.de (T.W.); 4Institute for Pathology, Otto-von-Guericke University Magdeburg, Leipziger Str. 44, 39120 Magdeburg, Germany; thomas.kalinski@med.ovgu.de

**Keywords:** pancreatic cancer, immunotherapy, IL-17A, IL-17RA, cancer stem cell

## Abstract

(1) Background: IL-17A accelerates pancreatic intraepithelial neoplasia (PanIN) progression. In this study, we examined whether IL-17A/IL-17RA promotes pancreatic ductal adenocarcinoma (PDAC) aggressiveness in terms of survival and cancer stem cell modulation. (2) Methods: In vitro, the wound-healing assay, the sphere formation assay, and flow cytometry were applied to assess cancer stem cell features. In vivo, pancreatic tumors were induced in C57BL/6 mice using electroporation with oncogenic plasmids (P53-/- R172H; KrasG12V). Anti-IL-17 antibodies were administered as immunotherapy. We analyzed IL-17A/IL-17RA related survival using publicly available transcriptomic data (*n* = 903). (3) Results: IL-17A/IL-17RA expression was not related to survival in PDAC patients. IL-17A neither induces stem cell markers nor increases sphere formation and cell motility in vitro. Blocking the IL-17A/IL-17RA axis in a murine pancreatic cancer model did not improve the survival of mice, but reduced the tumor burden slightly. (4) Conclusions: IL-17A does not promote stem cell expansion in PDAC cell lines. Blocking IL-17A/IL-17RA signaling does not interfere with pancreatic cancer development and progression and may not be considered as a promising monotherapy for PDAC.

## 1. Introduction

New therapies are urgently needed to overcome the dismal prognosis of pancreatic cancer. In other cancer entities such as malignant melanoma, immunotherapy improved the outcome substantially. Pancreatic intraepithelial neoplasia (PanIN) is an important precursor lesion for pancreatic cancer. McAllister et al. demonstrated that IL-17A establishes an immune to tumor cell crosstalk [1]. Kras mutated PanIN cells attract IL-17A producing immune cells, that in turn, promote PanIN proliferation. Thus, IL-17A produced by Th17 and γδ T cells establishes an immune system to tumor crosstalk via the IL-17A receptor on PanIN cells. The IL-17A accelerated initiation and progression of PanIN can be reversed by blocking IL-17A with antibodies. It has been suggested that clinical-grade neutralizing antibodies of IL-17 may represent an option for PanIN treatment [1]. It is unclear whether this observation is also valid for pancreatic cancer, and whether IL-17A/IL-17RA neutralization is a promising therapy for PDAC. Here, we ask whether IL-17A promotes the aggressiveness of pancreatic cancer both in vitro and in vivo.

## 2. Results

### 2.1. High IL-17A and High IL-17RA Expression Does Not Correlate with Overall Survival in Pancreatic Ductal Adenocarcinoma (PDAC) Patients

Human and mouse PDAC express IL-17RA (shown in Figure 1 and Appendix A). We detected varied expression levels in human PDAC within the tumor in nine patients. We also detected IL-17RA expression to a lower extent in the corresponding non-tumor tissue (Figure 1a). IL-17A stimulated human pancreatic cell lines did not show a dramatic difference in IL-17RA expression compared to naive cells (Figure 1b). We retrieved IL-17RA and IL17A gene expression levels from publicly available transcriptomic data of 903 patients with pancreatic cancer (Appendix A). The mRNA expression of IL-17RA and IL-17R in PDAC patients is not associated with the overall survival in this large cohort (p = 0.067 (IL17-RA), p = 0.261 (IL-17A), see Appendix A).

### 2.2. Treatment with IL-17A Does Not Induce Stem Cell Features in Pancreatic Cancer Cells In Vitro

Pancreatic cancer stem cells do exist and express specific markers including CXCR4, ABCG2, and CD44 [2,3]. Treatment of pancreatic cancer cell lines MIA PaCa-2 and Panc02 with human respective mouse recombinant IL-17A protein (20 ng/mL) did not change cell motility in the wound healing assay after 24 and 48 h (Figure 2a). Adding human recombinant IL-17A protein (20 ng/mL) to the human L3.6pl cell line does not increase sphere formation after seven days (1.05% vs. 1.02%, see Figure 2b). IL-17A did not increase CXCR4 (68.19% vs. 64.54%) and ABCG2 (1.96% vs. 1.52%) expression in human L3.6pl cancer cells assessed by flow cytometry. CD44 expression was not different in IL-17A treated Panc02 murine pancreatic cancer cells compared to the controls (1.73% vs. 1.38%, Figure 2c).

### 2.3. Blockade of IL-17 with Antibodies Does Not Improve Survival in a Mouse Model of Pancreatic Cancer

To investigate the IL-17A/IL-17RA axis in pancreatic cancer, we inhibited IL-17A signaling in a murine model of pancreatic cancer (Figure 3a). Treatment with a combination of anti-IL-17 antibodies did not improve the survival of experimental animals (Figure 3b). Survival in the treatment group was 44 ± 2.3 days on average in comparison with 47 ± 3.4 days in the control group. However, we observed two long-term survivors only in the anti-IL-17 treatment group, which was defined as animals surviving longer than 95 days.

There was a trend toward lower tumor weights in the anti-IL-17 treatment group on day 35 (Figure 3c, left) and on the day when the animals died (Figure 3c, right). However, no statistical significance was reached on day 35 when comparing control animals with animals receiving anti-IL-17 antibodies (*p* = 0.0529). At the time when animals died, there was also no statistically significant difference between the tumor weights in the control and anti-IL-17 group (*p* = 0.0504).

## 3. Discussion

Pancreatic cancer features a highly tolerant, “immune quiescent” tumor microenvironment because it lacks abundant infiltration of CD8+ effector T cells. Classic single-agent checkpoint inhibition accomplished no convincing responses in pancreatic cancer [4]. This finding indicates that the immune evasion mechanisms of pancreatic cancer are elaborated, and that effective immune therapy is hard to achieve [5]. Still, restoring the normal immune response and reversing harmful immune alterations is a principal aim when developing novel therapies for pancreatic cancer. It has been shown that IL-17A accelerates PanIN progression through crosstalk of immune cells with tumor cells [1]. If this mechanism functions in invasive pancreatic cancer, blocking IL-17A could be a promising therapy to target the altered immune response. In the current study, we found no survival benefit for IL-17A and IL-17RA low expressing tumors in 903 patients (Appendix A). Together, these results suggest that IL-17A/IL-17RA expression does not affect the survival of pancreatic cancer patients.

Cancer stem cells have the ability to initiate new tumors. They are resistant to chemotherapeutic treatments and thus might be responsible for tumor recurrence after cancer therapy. Targeting cancer stem cells is a major goal in immunotherapy. IL-17A has been shown to induce stem cell features in pancreatic cancer precursor PanIN cells [6]. However, we found no increased cell motility and sphere formation when treating pancreatic cancer cells with recombinant IL-17A (Figure 2). The addition of IL-17A to pancreatic cancer cells did not increase the subpopulation of cells expressing the typical stem cell markers such as CXCR4, ABCG2, and CD44. In contrast to PanIN cells, stem cell features in pancreatic cancer cells seems to be independent of IL-17A. While we have shown IL-17RA expression in both human and murine pancreatic cancer cells, IL-17A does not increase “aggressiveness” of pancreatic cancer cell lines in terms of inducing cancer stem cell features.

As IL-17A accelerates PanIN progression and PanIN is a precursor lesion of pancreatic cancer, we hypothesized that IL-17A might drive pancreatic cancer initiation and early progression. To assess the effect of IL-17A during pancreatic cancer development, we blocked IL-17A signaling with monoclonal antibodies in a murine pancreatic cancer model. This model features the induction of characteristic mutations that are locally restricted to the pancreatic parenchyma. In mice, these tumors are poorly differentiated, express cytokeratin 19, and clinically closely resemble the human disease. To assess the effect of IL-17 inhibition during tumor initiation and early tumor development, we started anti-IL17 therapy even before tumor induction. We injected the antibodies weekly to ensure IL-17A/IL-17RA axis inhibition throughout tumor development and progression. However, treatment with anti-IL-17 antibodies did not prolong overall survival in this animal model (Figure 3). We conclude that IL-17A may have different effects in PanIN and PDAC progression. Blocking the IL-17A/RA axis via antibodies alone is not able to effectively stop pancreatic cancer development and progression. Thus, this therapy is not suitable as a primary monotherapy for pancreatic cancer in clinical practice. We have asked whether the IL-17A/IL-17RA axis promotes the aggressiveness of pancreatic cancer. Our current in vitro and in vivo data could not show an association between IL-17A/IL-17RA signaling and the aggressiveness of pancreatic cancer.

Although statistical significance was not shown, there was a trend toward a lower tumor burden in the anti-IL-17 group early and late in the course of the disease (measured as weight of the tumor together with the pancreas). Interestingly, there were two long-term survivors in the anti-IL-17 group. The heterogeneity of the tumor cells and tumor stroma in PDAC might be a reason why only two animals benefited from IL-17 neutralization. The question is whether these long-term survivors were observed by coincidence or whether there are specific subgroups that benefit from IL-17A neutralization. To determine this, studies with larger groups of mice, different PDAC models, and more treatment styles might be necessary for future experiments.

## 4. Materials and Methods

### 4.1. Publicly Available Data Analysis

To analyze publicly available transcriptomic data, we downloaded the MetaGxData project from CodeOcean (https://codeocean.com/capsule/6438633/) published by Gendoo et al. [7] MetaGxData encompasses 11 manually curated datasets comprising RNA sequencing data together with standardized clinical, pathological, survival, and treatment metadata for pancreatic cancer. We established a local R repository installing the necessary libraries from Bioconductor’s ExperimentHub (https://bioconductor.org/). Finally, we adjusted the main R script from Gendoo et al. to calculate the survival of PDAC patients with available IL17A and IL17RA gene expression data. To calculate the survival curve for gene expression using the data of all 11 datasets, we split the patients in each dataset into two parts according to their respective gene expression. We analyzed the 33^th^ percentile as low expression and the 66^th^ percentile as high gene expression. Then, we combined the survival data of each dataset to estimate the survival of the combined 11 dataset.

### 4.2. Cells and Animals

The animal experiments complied with ethical regulations for animal research and was approved (reference number no.42502-2-1266 uniMD) by the Institutional Animal Care and Use Committee of the Magdeburg University Hospital. Animals were maintained, and experiments were conducted following institutional guidelines. Antibody (500 µg/mouse) i.p. anti-IL-17RA, anti-IL-17A, anti-IL-17F (1:1:1; kindly provided by Amgen, Germany), or PBS was applied to mice, according to the timetable (see Figure 2). We obtained MIA PaCa-2 pancreatic cancer cells from the American Type Culture Collection (ATCC, Manassas, VA). L3.6pl is a metastatic pancreatic cell line generated by our lab [8] and the Panc02 cell line is a murine pancreatic cancer cell line. Cells in culture were treated with recombinant human respective mouse IL-17A (20 ng/mL, Peprotech, Rocky Hill, CT, USA; Biolegend, San Diego, CA, USA) or PBS in the tests.

### 4.3. Pancreatic Cancer Animal Model

Pancreatic cancer was induced by in situ electroporation to transfect the pancreatic parenchyma with oncogenic plasmids in a locally restricted manner [9]. Engin Gürlevik generously provided transposon plasmids coding for KRAS-G12V, shRNA that silences p53, a gain-of-function mutant of p53, and a constitutively active version of Akt2 (generated by myristoylation of its N-terminus-myrAkt2). These four oncogenic plasmids were injected into the tail of the pancreas. Simultaneous co-delivery of the SB13-transposase achieved somatic integration of the transposon plasmids into the genome. The electroporation reproduces characteristic mutations of human PDAC. As a result, the clinical course of the model is very similar to the human disease. The application of the plasmids does not lead to increased induction of cancer stem cells. The emerging tumors showed local infiltration and developed distant metastases in the peritoneum, the liver, and the lungs. For the procedure, mice were anesthetized, and the pancreatic tail was mobilized after laparotomy. Plasmid DNA (50 μL of 0.5 μg/μL) was injected into the tail of the pancreas using a 27/30-gauge needle. Using an Electroporator (NepaGene), four electric pulses were administered twice (duration: 35 ms at 35 V, interval between pulses: 500 ms). The peritoneal cavity was washed and closed by suturing.

### 4.4. Sphere-Formation Assays

To generate tumor spheres, pancreatic cancer cells (3000 cells/well) were resuspended in serum-free DMEM/F12 medium (Thermo Fisher, Waltham, MA, USA) containing epidermal growth factor (20 ng/mL) and basic fibroblast growth factor (10 ng/mL) (Peprotech, Rocky Hill, CT, USA). Cells were seeded in 6-well ultra-low attachment plates. On day 7, the spheres that formed were counted by using a phase-contrast microscope (LEICA, Wetzlar, Germany).

### 4.5. Wound-Healing Assay

Pancreatic cancer cells were seeded in six-well plates (0.2 million cells/well). A micropipette tip was used to create a scratch wound when cells reached confluency. The size of the wound area was evaluated at 0, 24, and 48 h thereafter. The area of each wound was calculated with Image lab software.

### 4.6. Immunohistochemical Staining (IHC)

Tumors were fixed in formalin and embedded in paraffin before preparing 4-μm sections. For immunohistochemical analysis, sections were deparaffinized, rehydrated, and subjected to antigen retrieval for 30 min in 10 mM citrate buffer (pH 6) at 95 °C. After antigen retrieval, slides were blocked in Tris-buffered saline solution (pH 7.4) supplemented with 0.1% Tween-20 and 10% goat or donkey serum before incubation with primary antibodies overnight at 4 °C. Slides were washed and incubated with biotinylated anti-rabbit antibody and streptavidin-conjugated horseradish peroxidase for 30 min each. The staining was visualized using diaminobenzidine (DAB).

### 4.7. Flow Cytometry Analysis

One million cells were suspended in 100 μL PBS and incubated with PE-labeled anti-human CXCR4 (12G5, biolegend, 5 μL/million), anti-ABCG2 antibody (5D3, biolegend, 5 μL/million), respective anti-CD44 antibody (BJ18, biolegend, 5 μL/million) for 30 min on ice in the dark. Then, cells were washed and resuspended in 500 μL PBS. Cell counts were acquired on a BD flow cytometry system. FlowJo v10 software was used for data analysis.

### 4.8. Statistics

Difference between two groups was evaluated by the Student t-test. Data were considered to be significantly different with *p* < 0.05 in the two-tailed test. Error bars were shown as the mean ± SEM. Cumulative survival time was calculated with the Kaplan–Meier method and log-rank test using the R programming language. The R programming language and GraphPad Prism 6.0 was used to generate the images.

## Figures and Tables

**Figure 1 ijms-21-02215-f001:**
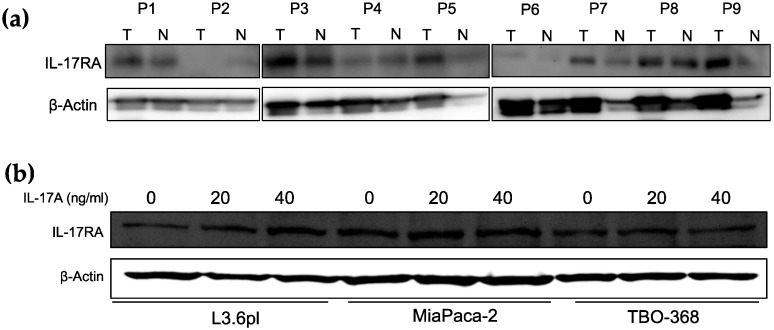
Analysis of IL-17A expression and survival of IL-17A high and low expressing tumors. (**a**) Western blot of IL-17RA protein expression in tumor (T) and corresponding non-tumor tissue (N) from nine human PDAC samples (P1–P9). (**b**) Western blot of IL-17RA protein expression in human PDAC cell lines L3.6pl, MIA PaCa-2, and patient derived PDAC cell line TBO368 with and without IL-17A stimulation (0, 20, 40 ng/mL) for 48 h.

**Figure 2 ijms-21-02215-f002:**
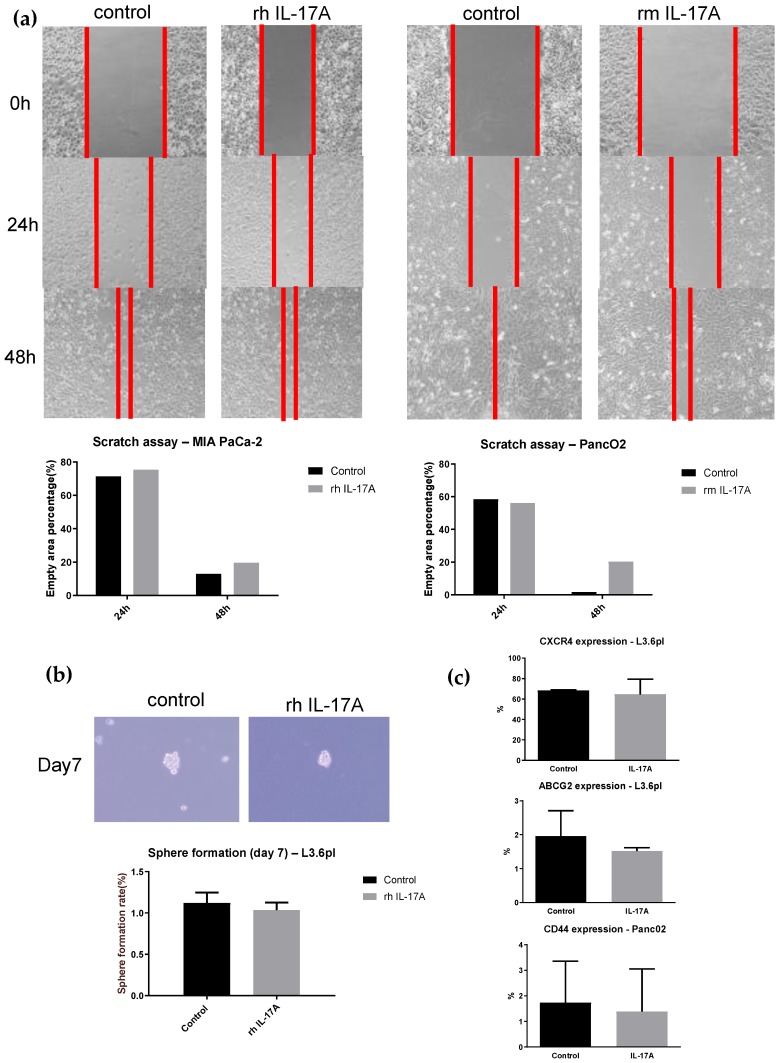
IL-17A does not induce cancer stem cell features in human or mouse PDAC cells *in vitro*. (**a**) Wound healing assay in pancreatic cancer cell lines. MIA PaCa-2 and Panc02 were treated 48 h with 20 ng/mL recombinant IL-17A or PBS. The wound area was assessed 24 and 48 h after scratching. The boundary of cancer cell migration is marked with red lines. (**b**) Sphere formation assay using human L3.6pl cells treated with 20 ng/mL recombinant human IL-17A or PBS for 48 h. Sphere formation rate at day 7 was calculated as the sphere number divided by the initial cell number plated and expressed as a percentage (Mean ± SEM). (**c**) Detection of human pancreatic stem cell markers ABCG2 and CXCR4 using flow cytometry in human pancreatic L3.6pl cells treated with 20 ng/mL recombinant human IL-17A or PBS for 48 h. CD44 was assessed in the pancreatic mouse cell line Panc02 under the same experimental setup.

**Figure 3 ijms-21-02215-f003:**
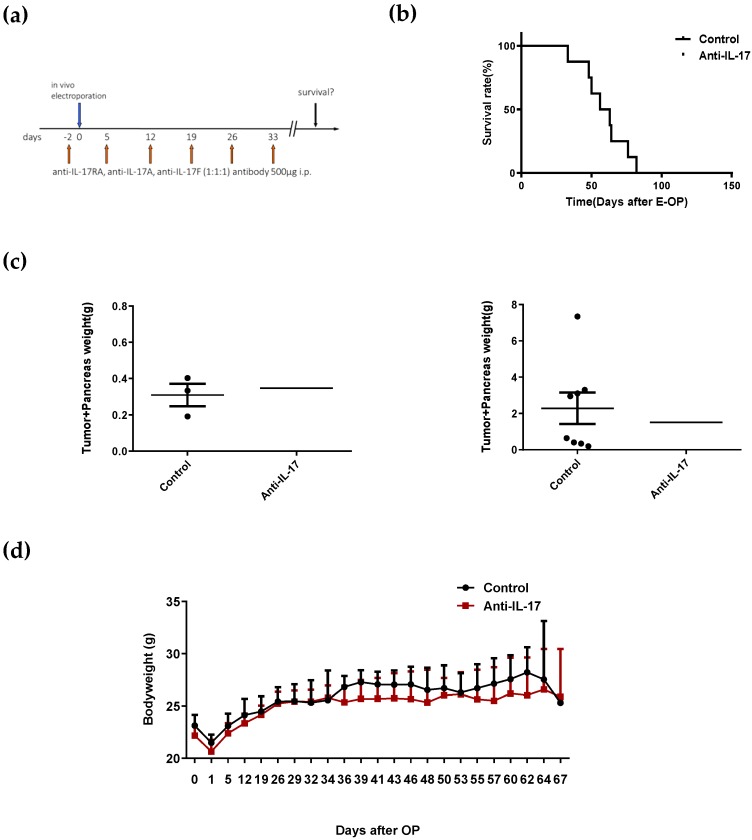
Blocking IL-17 does not affect the survival in a mouse model of PDAC. (**a**) Experimental setup: Animals received anti-IL-17 antibodies on days −2, 5, 12, 19, 26, and 33. In vivo electroporation to induce pancreatic cancer was performed on day 0. (**b**) Mice survival (%) of the control group (*n* = 8) vs. anti-IL-17 treatment group (*n* = 11). (**c**) Tumor weights (including the weight of the pancreas) on day 35 (*n* = 3, left) and endpoint (*n* = 6, right). (**d**) The curve of body weight from the control and anti-IL-17 treatment group was recorded to monitor the potential side effect.

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
