# Peer review of "The IL-17A/IL-17RA Axis Is Not Related to Overall Survival and Cancer Stem Cell Modulation in Pancreatic Cancer"

_ijms, 2020, doi:10.3390/ijms21062215_

Round 1
Reviewer 1 Report
This manuscript is the revised version of ijms-596149. Although the authors showed the expression of IL-17RA in PDAC samples (Fig S1 and Fig. 1A), the reviewer does not agree with the authors' opinion. Do the PADC cell lines, such as MiA PaCa-2, PancO2, and L3.6pl, express IL-17RA? Even if the cell lines express IL-17RA, IL-17A does not induce stem cell features or inhibit cell growth (Fig. 2). The authors emphasized the two long-termed survivors of the IL-17 signaling-inhibited group in a mouse model of pancreatic cancer (Fig 3); however, the reviewer wonder if the result can be reproducible. The reviewer does not agree with the authors' speculation in the lines 165-169 in the Discussion section without the reproducibility of appearance of long-termed survivors . The immunotherapy targeted to IL-17A/IL-17RA axis in PDAC might have some effect as anticancer therapy of PDAC with other anticancer strategies. But no results in the manuscript support the authors' conclusion/speculation.
Author Response
Dear Tammy, Dear Reviewers,
Thank you very much for your email from February 26th. The reviewers’ comments were again very helpful to strengthen the manuscript.
Please find a point-by-point response to the reviewer comments below. A revised version of our manuscript is also enclosed. We hope that this improved manuscript will convince yourself and reviewer one and hope to hear from you in due course with a positive response.
With our best regards
Reviewer #1:
This manuscript is the revised version of ijms-596149. Although the authors showed the expression of IL-17RA in PDAC samples (Fig S1 and Fig. 1A), the reviewer does not agree with the authors' opinion. Do the PADC cell lines, such as MiA PaCa-2, PancO2, and L3.6pl, express IL-17RA?
Response: We have detected a various human pancreatic cancer cell lines express IL-17RA. We added a western blot to the supplementary figure 1b to show IL-17RA expression of the cell lines MIA PaCa-2, L3.6pl and a PDAC patient derived cell line TBO368. Besides, we also displayed a single cell RNA sequencing data generated from the publication Byrnes et al Nature communication, 2018 Lineage dynamics of murine pancreatic development at single cell resolution. From this database we would find a general expression of IL-17RA in the mouse pancreas (shown in PDF).
Even if the cell lines express IL-17RA, IL-17A does not induce stem cell features or inhibit cell growth (Fig. 2).
Response: Although pancreatic carcinoma cell lines express IL-17RA, IL-17A does not induce the expression of either IL-17RA or some characteristics of stem cell properties. Our observation complements that of McAllister et al., who demonstrated IL-17A mediated induction of stem cell properties in pancreatic intraepithelial neoplasia cells (PanIN) cells. We expected that not only the pancreatic carcinoma precursor cells but also the pancreatic carcinoma cells themselves react to IL-17A. And with our in vitro and in vivo exploration, we did not observe a direct evidence of IL-17A alone could either induce stem cell feature on PDAC cell lines or inhibit PDAC cell progression.
The authors emphasized the two long-termed survivors of the IL-17 signaling-inhibited group in a mouse model of pancreatic cancer (Fig 3); however, the reviewer wonder if the result can be reproducible. The reviewer does not agree with the authors' speculation in the lines 165-169 in the Discussion section without the reproducibility of appearance of long-termed survivors. The immunotherapy targeted to IL-17A/IL-17RA axis in PDAC might have some effect as anticancer therapy of PDAC with other anticancer strategies. But no results in the manuscript support the authors' conclusion/speculation.
Response: we appreciated the argument from the Reviewer on the two long term survivors of the treatment group. We could not ignore the heterogenicity of the PDAC itself not only in human cases but also in the oncogenic plasmid induced mouse model. In the animal experiment set up, we have 8 mice in the control group and 11 mice in the treatment group. Therefore, we believe the reproducible of the result and we make more critical discussion for the two long-termed survivors in the manuscript.
We understand the speculation from the reviewer since there is very limit evidence of IL-17A/RA based therapy in PDAC. We believe, however, that it is justified in the discussion. IL-17A may work in combination with chemotherapy. Our work would encourage the scientific community to think more about whether further therapy attempts with IL-17A are useful. Only in the presented model, IL-17A is not as effective as we expected but it is how the scientific work developed. We could not only show the perfect experimental data but also display the negative observations for better communications. Nevertheless, we consider it to give interesting points and inspirations for the scientific community.

Reviewer 2 Report
The manuscript was adjusted according to the reviewers' comments and can be published in its current form.
Author Response
Dear Reviewer,
thank you very much for your support on our work.
With our kindest regards,
Yue Zhao and Felix Popp
Round 2
Reviewer 1 Report
In the revised manuscript, the authors showed the protein expression of IL-17RA in PDAC cell lines. But the IL-17A/IL-17RA signaling pathway seems to be inactive in these cell lines. Therefore, the IL-17A/IL-17RA axis does not involve PDAC aggressiveness. It is the conclusion to the purpose of the research. The purpose of the study is written in line 21 in the manuscript. In the reviewer's view, the finding that IL-17A via RORgt is associated with improved survival in pancreatic cancer patients is unrelated to the primary purpose. This is the reason the reviewer still does not agree with the authors' conclusion. If the authors justify the conclusion that IL-17A may work in combination with chemotherapy, they should change the primary purpose of the study and clearly state the point in the Introduction section, so that the reading audience can understand the authors' intention and the relationship between the object and the conclusion of the study. Also, descriptions of the stem cell features of PanIN cells should be removed from the Abstract and the Introduction section. They should be added to the Discussion section briefly. Overall the manuscript should be thoroughly revised so that the primary conclusion can be consistent with the primary objective.
Author Response
Dear Tammy, Dear Editorial team,
Thank you very much for your email. The reviewers’ comments were again very helpful to strengthen the quality of our manuscript and improved a lot of our current work.
We attached point-by-point response to the reviewer comments for your review. A revised version of our manuscript is also enclosed. We hope that this improved manuscript will convince you and the reviewer.
We are looking forward to hearing from you in due course with a positive response.
Million thanks for the support and concern,
With our best regards
Yue Zhao and Felix Popp
Reviewer #1:
In the revised manuscript, the authors showed the protein expression of IL-17RA in PDAC cell lines. But the IL-17A/IL-17RA signaling pathway seems to be inactive in these cell lines. Therefore, the IL-17A/IL-17RA axis does not involve PDAC aggressiveness. It is the conclusion to the purpose of the research. The purpose of the study is written in line 21 in the manuscript.
Response: We appreciated the statement from the reviewer that the IL-17A/IL-17RA axis does not affect PDAC aggressiveness. We substantially changed the manuscript to make this statement clear and to make the primary conclusion consistent with the primary objective.
In the reviewer's view, the finding that IL-17A via RORgt is associated with improved survival in pancreatic cancer patients is unrelated to the primary purpose. This is the reason the reviewer still does not agree with the authors' conclusion.
Response: The reviewer gave clear advice concerning the re-structure of our manuscript. Therefore we removed the RORgt finding and consequently changed the title to clarify the conclusion and the purpose of the research. The details have been marked in the main context of the manuscript.
If the authors justify the conclusion that IL-17A may work in combination with chemotherapy, they should change the primary purpose of the study and clearly state the point in the Introduction section, so that the reading audience can understand the authors' intention and the relationship between the object and the conclusion of the study.
Response: We would like to stick to the primary purpose of the study. As the reviewer initially suggested, we withdraw the statement that IL-17A blockade could work in combination with chemotherapy. Thus, we have made substantial changes to the discussion.
Also, descriptions of the stem cell features of PanIN cells should be removed from the Abstract and the Introduction section. They should be added to the Discussion section briefly.
Response: We removed the descriptions of the stem cell features of PanIN cells from the abstract and the introduction section. Indeed, we should not make too much focus on stem cell features of PanIN in this study, since we are major focus on the PDAC.
Overall the manuscript should be thoroughly revised so that the primary conclusion can be consistent with the primary objective.
Response: We appreciated the advice from the reviewer and thoroughly revised the manuscript. We made substantial changes to the abstract and the discussion, as described above. Also, we have almost completely rewritten the introduction. And we hope the update version could achieve the requirement of the short communication so that we could expect more readers to know and get the pre-clinical therapeutic experience of IL-17A/IL-17RA axis in PDAC.
Round 3
Reviewer 1 Report
In the second revised manuscript, the authors responded appropriately to the reviewer’s comments. The reviewer now understands what the authors intend their findings for reading audiences. However, the reviewer still wants the authors to respond to the following comments for further improvement of the manuscript.
1. Lines 20-21, the sentence, “If IL-17A accelerates~.”, can be removed.
2. Lines 42-43, The IL-17A accelerated initiation and progression of PanIN, which can be reversed by blocking IL-17A with antibodies. Is it OK to add some words indicated by underline?
3. Line 45, Here, we investigate whether IL-17A promotes the aggressiveness of PDAC both in vivo and in vitro. Is it OK to modify the sentence indicated by underline?
4. Lines 46-48, the sentences, “On the other hand, we ~ tumor recurrence later on [2].”, can be removed.
5. Lines 48-52, the sentences, “ Pancreatic cancer stem cells ~ IL-17A promoting PanIN progression”, can be moved to the Results section 2.2.
6. Lines 52-54, the sentence, “It could be ~ PDAC compared to PanIN.” can be removed, or moved to the Discussion section if necessary.
7. Figure 1 (C) can be included in Figure S1.
8. Figure S1 should be removed from the manuscript. It should be added as a supplemental file.
9. Lines 126-127, add “(Figure S1)” at the bottom of the sentence.
10. Lines 132-133, add “(Figure 2)” at the bottom of the sentence.
11. Lines 146-147, the sentence, “However, treatment with ~ in this animal model.”, can be removed.
12. Line 147, the word, “However”, can be removed.
13. Lines 150-151, add “(Figure 3)” at the bottom of the sentence.
Author Response
Dear Tammy, Dear Editorial team,
Thank you very much for your email. We appreciated a lot for the reviewers’ comments to improve our current manuscript.
We attached point-by-point response to the reviewer comments for your review. A revised version of our manuscript is also enclosed.
We hope that this improved manuscript will convince you and the reviewer for a positive consideration.
Million thanks for the support and concern,
With our best regards
Yue Zhao and Felix Popp
Response
In the second revised manuscript, the authors responded appropriately to the reviewer’s comments. The reviewer now understands what the authors intend their findings for reading audiences. However, the reviewer still wants the authors to respond to the following comments for further improvement of the manuscript.
- Lines 20-21, the sentence, “If IL-17A accelerates~.”, can be removed.
Response: thank you for the comments and we removed this sentence.
- Lines 42-43, The IL-17A accelerated initiation and progression of PanIN, which can be reversed by blocking IL-17A with antibodies. Is it OK to add some words indicated by underline?
Response:we add additional sentence to strength the expression: It has been suggested that clinical-grade neutralizing antibodies of IL-17 may represent an option for pancreatic cancer prevention and/or treatment.
- Line 45, Here, we investigate whether IL-17A promotes the aggressiveness of PDAC both in vivo and in vitro. Is it OK to modify the sentence indicated by underline?
Response:we modified as the reviewer has suggested in the revised version.
- Lines 46-48, the sentences, “On the other hand, we ~ tumor recurrence later on [2].”, can be removed.
Response:we removed the part as the reviewer has suggested in the revised version.
- Lines 48-52, the sentences, “Pancreatic cancer stem cells ~ IL-17A promoting PanIN progression”, can be moved to the Results section 2.2.
Response:We moved the sentences to the Results section 2.2. as reviewer suggested and deleted some context for better correlation with the results.
- Lines 52-54, the sentence, “It could be ~ PDAC compared to PanIN.” can be removed, or moved to the Discussion section if necessary.
Response: we removed the part as the reviewer has suggested in the revised version.
- Figure 1 (C) can be included in Figure S1.
Response: we changed Figure 1 (C) into Figure S1 as reviewer suggested.
- Figure S1 should be removed from the manuscript. It should be added as a supplemental file.
Response:we removed the S1 figure into supplemental file as reviewer suggested.
- Lines 126-127, add “(Figure S1)” at the bottom of the sentence.
Response:we have add the Figure S1 at the bottom of the sentence.
- Lines 132-133, add “(Figure 2)” at the bottom of the sentence.
Response:we have add the Figure 2 at the bottom of the sentence as suggested.
- Lines 146-147, the sentence, “However, treatment with ~ in this animal model.”, can be removed.
Response:we could not match the context of ‘However, treatment with ~ in this animal model’. But we removed the sentence of ‘However, the current PDAC mouse model is not suitable to analyze stem cell features in vivo directly.’ for review.
- Line 147, the word, “However”, can be removed.
Response: we have removed the word and the new location of the sentence is in line 137 in the revised version.
- Lines 150-151, add “(Figure 3)” at the bottom of the sentence.
Response:we have add the Figure 3 at the bottom of the sentence as suggested.
Round 4
Reviewer 1 Report
The manuscript was adjusted according to the reviewers' comments.
This manuscript is a resubmission of an earlier submission. The following is a list of the peer review reports and author responses from that submission.
Round 1
Reviewer 1 Report
Li et al. investigated whether inhibition of IL-17A/IL-17RA axis suppresses tumor progression of pancreatic cancer cells. Unlike with the PanIN cells, IL-17A treatment does not lead cancer stem cell features in the PDAC cells in vitro (Figure 3). Further, in vivo model of pancreatic cancer in mice, treatment with a combination of anti-IL-17A, anti-IL-17F, and anti-IL-17RA antibodies did not improve survival of the experimental mice although the authors observed two long-termed survivors in the antibodies-treated group (Figure 2). Contrary to the authors’ expectation, the expression of RORC, the master transcription factor of IL-17A, in the pancreatic tumor samples is associated with better overall survival in patients (Figure 1). The authors conclude that the combination therapy with other anti-cancer strategies needs to be further evaluated (for the development of the novel and effective anticancer therapy).
In the reviewer’s opinion, the authors should have determined the expression of IL-17RA, the receptor for IL-17A, in pancreatic cancer cells at the beginning of the study. If they don’t express IL-17RA, there is not much point in blocking the IL-17A/IL-17RA axis to prevent the progression of pancreatic cancer cells directly.
The explanation of the data in figure 2 seems to be complex. The reviewer could not understand whether the authors focused on the induction of stem cell features or progression of pancreatic cancer cells. In the experiment, they induced pancreatic cancer cells by introduction of mutant RAS, mutant p53, and constitutively active Akt2 genes. Were the stem cell features induced in gene-transduced cells during pancreatic cancer formation?
In Figure 1B, the reviewer does not think pancreatic cancer cells were stained with an anti-RORγt antibody. Instead, some immune cells such as Th17 in the pancreatic tumor could be stained, as RORγt is a key transcription factor mediating Th17 cell differentiation as well as IL-17 production. Further, RORγt is the T cell-specific isoform of RORγ and it controls gene networks that enhance immunity including increased IL-17 production and decreased immune suppression (PMID: 26785144). Even if the RORC gene was induced in pancreatic cancer cells, its expression may negatively regulate tumor progression. Actually, RORC expression was associated with clinicopathologic features of bladder patients (PMID: 30808674). From above, it seems obvious that high RORC and high RORγt expression in tumor samples correlates with better overall survival.
Taken together, the reviewer does not agree with the authors’ conclusion/opinion in the current version of the manuscript.
Reviewer 2 Report
The paper describes the investigation of the blockade of IL-17A/IL-17RA axis in pancreatic cancer progression and the significance of RORgt expression and cancer stem cells modulation. The authors have conducted a sound scientific work, planning appropriate experiments and presenting their results in a succint manner. Results were statistically evaluated and support the conclusion formulated. It is my opinion that the paper fits the aim of the journal and can be published in its current form.